# Creating fluorescent quantum defects in carbon nanotubes using hypochlorite and light

Ching-Wei Lin [1], Sergei M. Bachilo [2], Yu Zheng[2], Uyanga Tsedev [1,3], Shengnan Huang [1,4], R. Bruce Weisman [2,5] & Angela M. Belcher [1,3,4]

Covalent doping of single-walled carbon nanotubes (SWCNTs) can modify their optical properties, enabling applications as single-photon emitters and bio-imaging agents. We report here a simple, quick, and controllable method for preparing oxygen-doped SWCNTs with desirable emission spectra. Aqueous nanotube dispersions are treated at room temperature with NaClO (bleach) and then UV-irradiated for less than one minute to achieve optimized O-doping. The doping efficiency is controlled by varying surfactant concentration and type, NaClO concentration, and irradiation dose. Photochemical action spectra indicate that doping involves reaction of SWCNT sidewalls with oxygen atoms formed by photolysis of ClO$^-$ ions. Variance spectroscopy of products reveals that most individual nanotubes in optimally treated samples show both pristine and doped emission. A continuous flow reactor is described that allows efficient preparation of milligram quantities of O-doped SWCNTs. Finally, we demonstrate a bio-imaging application that gives high contrast short-wavelength infrared fluorescence images of vasculature and lymphatic structures in mice injected with only ~100 ng of the doped nanotubes.

[1] The David H. Koch Institute for Integrative Cancer Research, Massachusetts Institute of Technology, Cambridge, MA 02139, USA. [2] Department of Chemistry and the Smalley-Curl Institute, Rice University, Houston, TX 77005, USA. [3] Department of Biological Engineering, Massachusetts Institute of Technology, Cambridge, MA 02139, USA. [4] Department of Materials Science and Engineering, Massachusetts Institute of Technology, Cambridge, MA 02139, USA. [5] Department of Materials Science and NanoEngineering, Rice University, Houston, TX 77005, USA. Correspondence and requests for materials should be addressed to R.B.W. (email: weisman@rice.edu) or to A.M.B. (email: belcher@mit.edu)

One of the most intriguing properties of semiconducting single-wall carbon nanotubes (SWCNTs) is their structure-specific fluorescence at short-wave infrared (SWIR) wavelengths[1,2]. This has inspired emerging applications in areas that include bioimaging[3–9] and optical noncontact sensing[10–12]. In addition, it has been shown that SWCNTs with some types of sparse covalent doping give spectrally shifted emission arising from the trapping of mobile excitons at the defect sites. Such intentionally doped nanotubes have been used to construct the first room temperature single-photon source emitting at telecom wavelengths[13–15], a key step for the development of quantum communications and cryptography[16,17]. The nanotube quantum defects are either ether-bridged oxygen atoms[18–20], which leave all carbon atoms $sp^2$-hybridized, or organic addends[21–23], which convert nanotube atoms from $sp^2$ to $sp^3$ hybridization at the functionalization sites. Besides the ether conformation, oxygen doping is also known to generate epoxide adducts, which are less stable than the ether-bridged structures[18,24]. The sparse energy traps resulting from doping apparently suppress fluorescence quenching by dark excitons or structural defects, thereby increasing the nanotube emissive quantum yields[18,21]. Unlike pristine SWCNTs, those with sparse doping show significant Stokes shifts between their SWIR absorption and emission bands. This property allows bioimaging with SWIR excitation, reducing excitation scattering and suppressing autofluorescence from biological tissues[18,20]. The fluorescent quantum defects also brighten ultrashort SWCNTs[25], which have potential biomedical advantages because of their size[26–28] but are otherwise nonemissive because of end quenching[25,29].

Broader use of SWCNTs containing fluorescent defects has been hampered by preparation methods that require special reactants, are difficult to control, proceed slowly, generate nonemissive defects, or are challenging to scale. We report here a simple, quick, and controllable way to efficiently generate oxygen-doped SWCNTs. Surfactant-suspended nanotubes in the presence of NaClO (bleach) are irradiated in the near-UV to induce photodissociation of $ClO^-$ and form the desired doped SWCNTs. The doping density is readily controlled by illumination time, with maximal defect emission intensity reached in less than 1 min. We characterize the reaction product by absorption, fluorescence, Raman, variance, and single particle spectroscopies and propose a simple reaction mechanism. We also describe a simple continuous flow reactor for efficiently preparing O-doped SWCNTs and demonstrate sensitive in vivo imaging in mice using SWIR fluorescence from our doped samples.

## Results

**Optical characterization of products**. Figure 1a shows the fluorescence spectra of bulk samples of (6,5)-sorted SWCNTs before and after treatment with bleach and UV light. Clear evidence of successful O-doping is the appearance of an intense red-shifted emission peak $\left(E_{11}^*\right)$ at 1126 nm and the decreased intensity of the pristine $E_{11}$ emission peak at 988 nm. The observed $E_{11}^*$ wavelength matches values reported for (6,5) O-SWCNTs by Ghosh et al.[18] and Chiu et al.[19] Treatment shifts most of the sample emission from the pristine band to $E_{11}^*$, indicating that a large fraction of nanotube excitons emit while trapped at O-doped sites. The long-wavelength side bands in the treated sample are assigned as $X_1^*$ transitions (see Supplementary Fig. 2) and/or emission from parallel epoxide defects[18,24]. We find that treatment increases the spectrally integrated emission by a factor of 2.6 (see Supplementary Fig. 2a). Figure 1b plots the sample's absorption spectra before and after doping treatment.

Peak absorbance drops by 17.2% at $E_{11}$ and by 6.9% at $E_{22}$ after treatment. We attribute these changes to perturbations in the $\pi$-electron system from covalent doping. The inset in Fig. 1b, showing absorbance on a magnified scale, reveals for the first time a new induced feature peaking at 1114 nm, with an absorbance value ~1.6% of the pristine $E_{11}$ value. We assign this to $E_{11}^*$ absorption, analogous to the observed weak absorption band reported in $sp^3$–doped SWCNTs[21].

Figure 1c shows the excitation–emission profiles of pristine and O-doped SWCNT samples. Treatment shifts the coordinates of the dominant feature from (565 and 988 nm) to (988 and 1126 nm). Raman D/G band intensity ratios are commonly used to monitor covalent sidewall functionalization in SWCNTs. As shown in Fig. 1d, the D/G ratio of our sample increased from 0.013 to 0.037 on doping treatment. This final D/G ratio is notably smaller than values reported using other methods for generating fluorescent quantum defects in SWCNTs, suggesting minimal nonemissive defects from side reactions.

**Reaction investigations**. Fluorescence spectroscopy is the preferred method for observing the conversion of pristine to O-doped SWCNTs. Fortunately, in our reaction it is possible to use a single ultraviolet light source both to induce the reaction with $ClO^-$ and also to excite sample fluorescence to monitor the extent of product formation. Figure 1e shows time dependent intensities of the $E_{11}$ and $E_{11}^*$ emission peaks from a dispersed (6,5)-sorted sample in aqueous sodium cholate (SC) and NaClO as it is irradiated with a few milliwatts of 300 nm light. This in situ monitoring reveals a clear maximum in $E_{11}^*$ intensity near 40 s while the pristine emission decays monotonically. Separate spectral measurements of the $ClO^-$ concentration show that it decreases to less than 5% of its initial value by ca. 40 s (see Supplementary Fig. 8), indicating that the reactant is almost fully consumed in the first minute of irradiation.

To investigate the reactions tracked in Fig. 1e, we performed two control experiments. In the first, SWCNT fluorescence was measured after irradiation at 300 nm for 50 s in the absence of NaClO (Fig. 1f). We found very little change in the sample's emission intensity or spectral shape, indicating that NaClO is essential for the reaction. In the second control experiment, samples contained NaClO but were not exposed to UV light. Here a suspension of (6,5)-enriched SWCNTs in 0.035% SC/0.75 mM NaClO was split into two aliquots. Aqueous 0.1% sodium deoxycholate (SDC) was immediately added to the first aliquot to protect the nanotubes from reaction and provide a reference. The other aliquot was kept in dark for 24 h before SDC was added. Spectral changes between the two portions were then measured and quantified. Exposure to NaClO for 24 h in the dark led to no significant $E_{11}^*$ feature or increase in Raman D/G ratio. This agrees with the result reported by Chiu et al.[30], who used an enzyme to produce low concentrations of $ClO^-$ ions. We did observe a 39% decrease in $E_{11}$ emission and a broad reduction in absorption, probably reflecting some oxidative destruction of SWCNTs by $ClO^-$ (see Supplementary Fig. 9). However, that process is negligibly slow on the scale of our sub-minute reaction time. We conclude that both $ClO^-$ and photoexcitation are required for the new O-doping reaction.

A key clue to a photochemical reaction's mechanism is its action spectrum, which we investigated by measuring fluorescence changes in replicate samples irradiated at various wavelengths. Figure 2a shows the relative increase in $E_{11}^*$ emission after treatment with irradiation at the first, second, and third SWCNT resonant absorption bands, as well as at 300 nm. The results have been normalized to irradiation power. They indicate that the doping reaction is not induced by direct SWCNT

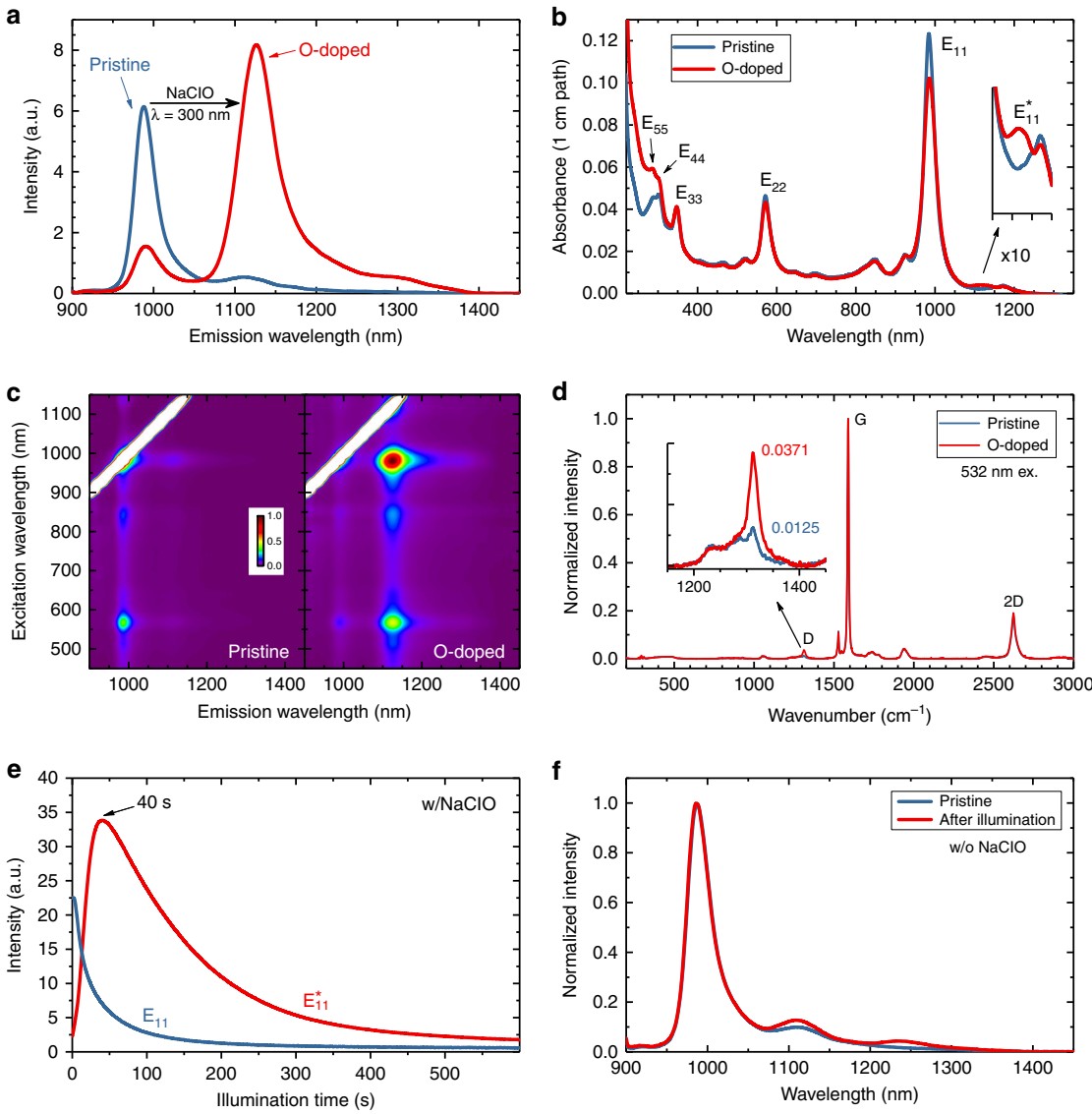

**Fig. 1** Optical properties of pristine and O-doped (6,5)-enriched SWCNT samples. **a** Fluorescence spectra. **b** Absorption spectra; expanded inset shows $E_{11}^*$ absorption feature. **c** Excitation-emission maps. **d** Raman spectra; expanded inset shows the D bands. **e** Emission intensities versus illumination time. **f** Normalized emission spectra of SWCNT samples with and without 300 nm irradiation in the absence of NaClO

excitation. In Fig. 2b, red circles show doping rates (corrected for irradiation power) at a number of UV wavelengths. The increase in rates at shorter irradiation wavelengths indicates that the doping reaction is aided by excess energy in the photogenerated reactant, which we deduce to be oxygen atoms formed through photodissociation of aqueous $ClO^-$ ions. A plausible channel for oxygen excitation would be its release in the excited $^1D$ state rather than the $^3P$ ground state, which can occur for $ClO^-$ irradiation wavelengths below 320 nm. The $^1D$ oxygen could then react with SWCNTs in a spin-allowed process. In Fig. 2b, the two "dark yellow" triangles show the relative formation rates of $^1D$ oxygen atoms, based on our experimental parameters and the photodissociative quantum yields reported by Buxton et al.[31,32] We propose the following general mechanism for photoinduced oxygen doping of SWCNT in the presence of aqueous NaClO:

$$ClO^- \xrightarrow{h\nu} O + Cl^- \qquad (1)$$

$$SWCNT + O \rightarrow SWCNT - O, \qquad (2)$$

giving the overall reaction

$$SWCNT + ClO^- \xrightarrow{h\nu} SWCNT - O + Cl^-. \qquad (3)$$

Figure 2c shows semiempirical quantum chemical energies for the reactant and product species in the proposed mechanism, which is illustrated in Fig. 2d. The major O-SWCNT product is the ether form rather than the epoxide. Doping using photolyzed $ClO^-$ may give higher selectivity towards the ether product as compared to the original ozone method (See Supplementary Table 4).

**Effects of surfactant and hypochlorite concentrations.** Surfactant concentration is an important parameter in the O-doping reaction, as can be seen from the pristine and shifted emission intensities plotted in Fig. 3a. Nanotube doping is minimal at high concentrations of SC and greatest below the critical micelle concentration (CMC) of 17 mM. We expect that high surfactant concentrations enable effective micellar shielding of nanotube surfaces from dissolved species, preventing reactions with photochemically generated oxygen atoms. This shielding effect is

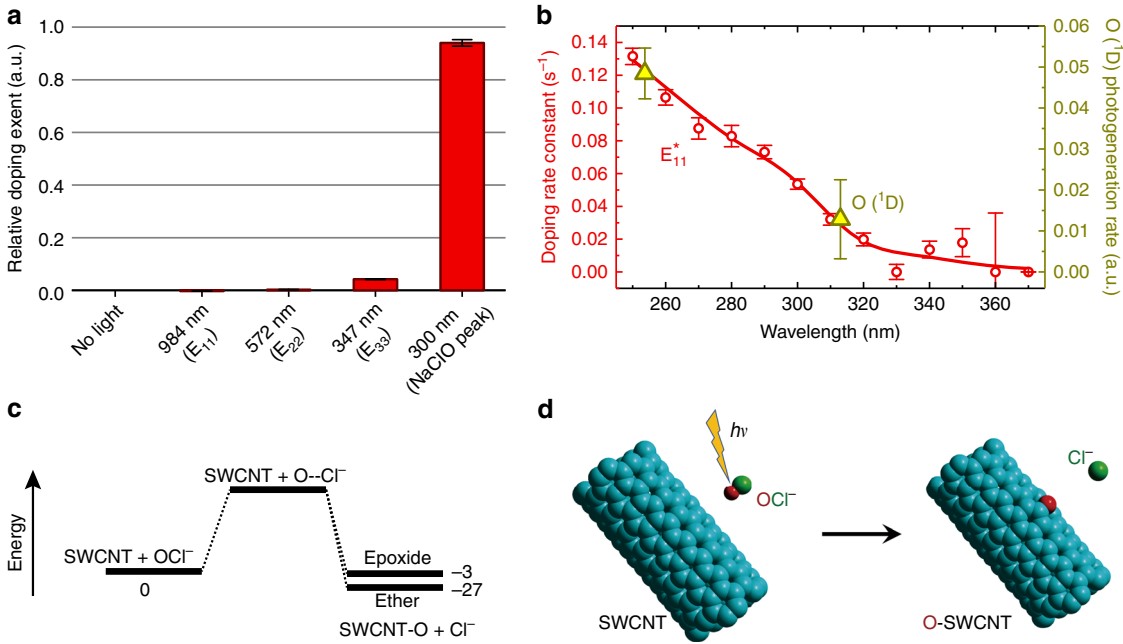

**Fig. 2** Mechanism of oxygen doping using NaClO. **a** The relative doping extents in samples irradiated at SWCNT resonant absorption peaks and at the 300 nm NaClO absorption peak. Relative doping extent is defined as $\left[\left(E_{11}^{*}/E_{11}\right)_{irrad.} - \left(E_{11}^{*}/E_{11}\right)_{no\ light}\right]/P_{irrad.}$. Error bars indicate standard deviation (s.d.) from the spectral measurement noise. **b** Action spectrum of the doping rate constant (circles), and relative photogeneration rates of O ($^1$D) (triangles). Error bars indicate s.d. estimated from literature uncertainties and spectral measurement noise. **c** Diagram showing computed energies for reactants, products, and proposed transition state for the doping reaction in vacuum (units: kcal mol$^{-1}$). Note that the ether adduct is more stable than the epoxide. **d** Illustration of the proposed doping reaction

similar to the strong coating dependence in the recently reported reversible quenching of SWCNT fluorescence by dissolved $O_2$[33]. Here, we find the optimal SC concentration for doping to be 0.035–0.07%, corresponding to 8–16 mM, or below the CMC of SC. Fortunately, nanotube aggregation at these low surfactant concentrations is not a concern on the short time scale of the doping reaction.

Figure 3b compares the maximum $E_{11}^{*}$–$E_{11}$ ratios obtained with four common nanotube surfactants: sodium dodecylbenzene sulfonate (SDBS), SDC, sodium dodecyl sulfate (SDS), and SC. They are shown in the order of their CMC values. Both SC and SDS give high doping reaction yields, consistent with their behavior as weaker agents for SWCNT dispersion. We note that unexcited NaClO quenches the fluorescence of SWCNTs in 2% w/ v SDS. We suspect that even at this high surfactant concentration, HClO in the slightly acidic solution can penetrate to the nanotube surface and strongly perturb the $\pi$-electron system. Subsequent addition of a competing surfactant such as SC or SDC restores the fluorescence. Both SDBS and SDC permit very little oxygen doping during our photoexcited bleach treatment, even when the SDC concentration is tuned to below $3 \times 10^{-3}$% and $E_{11}$ emission is too weak to be observed. We conclude that surfactant identity and concentration are important parameters that control access of oxygen doping reactants to the nanotube surface.

Figure 3c plots the emission intensities of treated SWCNTs as a function of NaClO concentration. The strongest doping is found near 0.1 mM, corresponding to a ratio of NaClO molecules to carbon atoms of ~3. We suggest that lower NaClO concentrations generate suboptimal densities of oxygen doping sites, whereas higher concentrations lead to excessive nonemissive defects, lowering both the pristine and doped emission intensities (see Supplementary Fig. 13 for D/G Raman ratios).

**Doping analysis**. The extent and homogeneity of O-doping in treated nanotubes is important for applications such as

fluorescent probes and single-photon sources. To characterize these parameters, we supplemented ensemble spectral measurements with variance and single-particle emission spectroscopies. Variance spectroscopy[34–36] is a recently developed method that evaluates the statistical differences among many replicate emission spectra from small volumes of a dilute sample to find the concentrations and associations of various emitting species. Variance data from a sample can be plotted to a show a covariance contour map in which diagonal features represent emission peaks of distinct particles and off-diagonal features arise from particles that emit at two different wavelengths. Figure 4a, b shows such covariance maps for samples of pristine and O-doped (6,5)-SWCNTs (see Supplementary Fig. 14a for the corresponding mean spectra). The map for the pristine sample has a single diagonal feature at the 994 nm $E_{11}$ peak. After doping treatment, that feature becomes less intense and a dominant diagonal $E_{11}^{*}$ peak appears at 1126 nm. Figure 4c compares the two variance spectra (diagonal traces in the covariance maps) before and after doping. An interesting feature of the O-doped variance spectrum is barely resolved $E_{11}^{*}$ peaks ~6 nm apart. We assign these to emission from doped (6,5) SWCNTs of opposite helicity. It has been reported that such enantiomeric spectral shifts can be induced by differing coating structures of chiral cholate surfactants on SWCNTs[35,37,38]. The most significant qualitative finding from our variance data is the presence of clear off-diagonal features in Fig. 4b (marked by white arrows) that reveal strong spatial correlations between 994 nm and 1126 nm fluorescence. These demonstrate that both pristine and O-doped emissive sites coexist on individual nanotubes, in agreement with previous findings from a single-particle imaging study[39].

Quantitative analysis of covariance spectral data lets us estimate the relative populations of treated SWCNTs showing pristine and doped emission. This requires the Pearson correlation coefficient ($\rho$) for signals at the two peak positions, which

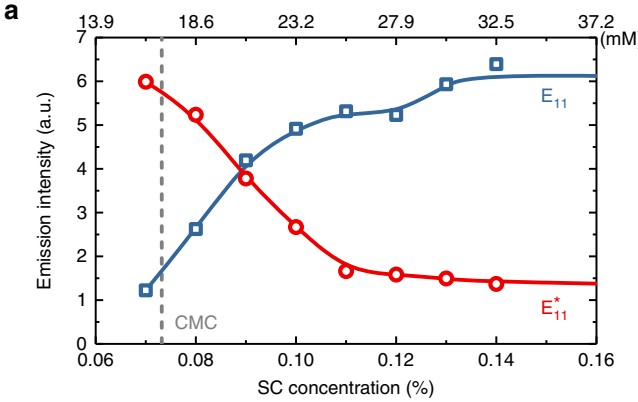

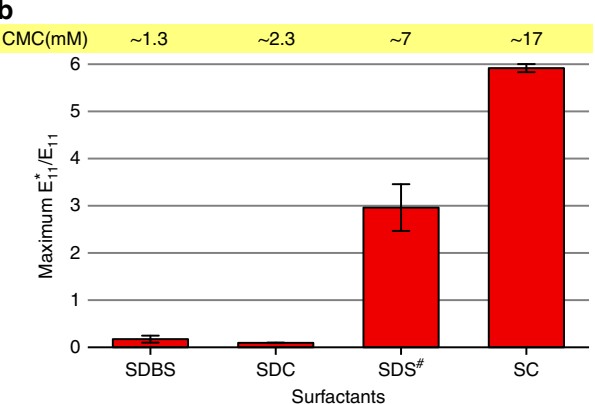

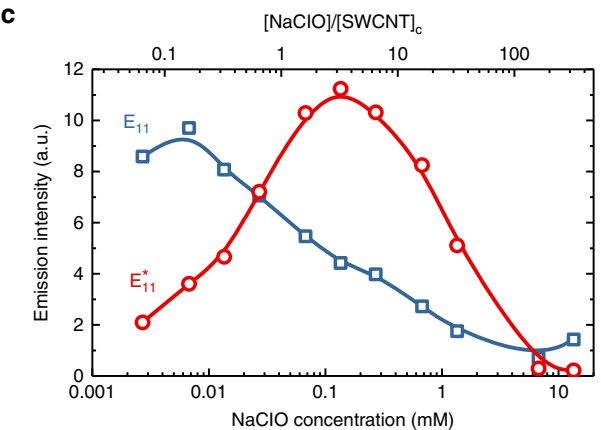

**Fig. 3** Effects of surfactants and reactant on doping efficiency. **a** $E_{11}$ and $E_{11}^*$ emission intensities after treatment of samples suspended in various SC concentrations. **b** Maximum ratio of $E_{11}$-$E_{11}^*$ emission intensities for treated samples suspended in different surfactants. The hash symbol represents SWCNT fluorescence quenched after addition of oxidizing agent. Error bars indicate s.d. from the spectral measurement noise. **c** NaClO concentration dependence of $E_{11}$ and $E_{11}^*$ emission intensities after treatment

can be expressed as

$$\rho_{\lambda_j}(\lambda_k) = \frac{\mathrm{cov}_{\lambda_j}(\lambda_k)}{e_{\lambda_j}(\lambda_k)} \qquad (4)$$

Here, $\mathrm{cov}_{\lambda_j}(\lambda_k)$ is the covariance at wavelengths $\lambda_j$ and $\lambda_k$ normalized to the variance at $\lambda_j$ (or $\mathrm{cov}(\lambda_j, \lambda_k)/\sigma^2(\lambda_j)$), and $e_{\lambda_j}(\lambda_k)$ is the emission efficiency at $\lambda_k$ normalized to the efficiency at $\lambda_j$. Figure 4d plots the full correlation spectra $\rho_{994\,\mathrm{nm}}(\lambda)$ and

$\rho_{1126\,\mathrm{nm}}(\lambda)$, which are horizontal traces through the covariance map at the 994 and 1126 peaks. The two plateaus near those $E_{11}$ and $E_{11}^*$ emission wavelengths represent strongly correlated components. Their magnitudes indicate that ~91% of the $E_{11}^*$ emitting SWCNTs also show $E_{11}$ emission while ~73% of the $E_{11}$ emitting SWCNTs also emit at the $E_{11}^*$ wavelength. We can further deduce the abundances of the following three categories of SWCNTs in the treated sample: ~26% remain undoped, showing $E_{11}$ emission only; ~7% show $E_{11}^*$ emission only; and ~67% show both $E_{11}^*$ and $E_{11}$ emission. Note that one cannot deduce these abundances just from the $E_{11}^*/E_{11}$ ratio, and the fraction of undoped SWCNTs may be less than 5% even when the $E_{11}^*/E_{11}$ peak ratio is only ~1.5 (see Supplementary Fig. 20).

Single-particle measurements reveal additional information about dopant homogeneity. As shown in Supplementary Fig. 22, we used spectrally filtered SWIR fluorescence microscopy to separately measure $E_{11}^*$ and $E_{11}$ emission from many individual SWCNTs in treated and control nanotube samples. We then examined the correlation of the $E_{11}^*/E_{11}$ ratio with the total particle emission, which is an approximate gauge of nanotube length. O-doped SWCNTs show a positive correlation between intensity ratio and nanotube length. This implies that doping is not restricted to sites at the nanotube ends, because that would lead to relatively more pristine emission in longer nanotubes and a negative correlation. This conclusion is consistent with previous findings based on fluorescence imaging of individual doped nanotubes[39]. We also find that longer SWCNTs in treated samples show less variation in $E_{11}^*/E_{11}$ ratio than shorter SWCNTs. We suggest that this is because the smaller average number of doping sites in short nanotubes leads to larger statistical fluctuations in their spectral signatures. Based on the recent study by Danné et al.[25], we expect that ultrashort O-doped SWCNTs are more likely to show only $E_{11}^*$ emission. Length heterogeneity in samples of O-doped SWCNTs therefore contributes to the observed spectral heterogeneity.

**High-throughput reactor for in vivo imaging**. We have constructed a custom-designed flow reactor for the efficient production of O-doped SWCNTs. Figure 5a schematically illustrates our device. NaClO solution and a concentrated SWCNT suspension (OD = 34 cm$^{-1}$ at $E_{11}$) are loaded into separate syringes and then mixed just before injection into a spectrophotometric flow cell used as the reaction chamber. The mixture is illuminated by light from a 300 nm LED, which induces the reaction and also excites fluorescence in the sample. The resulting nanotube emission is transmitted to a near-IR spectrometer for monitoring. Immediately following the doping reaction, we added extra SC surfactant to protect the SWCNT sidewalls and prevent possible aggregation or side reactions. Figure 5b plots the emission spectrum of a treated sample containing 6 mg mL$^{-1}$ of (6,5)-SWCNTs, as determined from its $E_{11}$ peak absorbance of ~3 cm$^{-1}$ and the known (6,5) absorptivity[40]. We estimate that our device can produce up to ca. 0.3 mg h$^{-1}$ of O-doped SWCNTs per mL of reaction chamber under 29 mW cm$^{-2}$ of UV illumination.

**In vivo imaging using O-doped SWCNTs**. As was discussed in a prior report, O-doped SWCNTs are preferable to pristine SWCNTs for bioimaging because their fluorescence can be excited at the $E_{11}$ transition and detected at $E_{11}^*$. The use of longer wavelength excitation allows better tissue penetration and greatly suppressed autofluorescence backgrounds. To demonstrate this application, we prepared a batch of O-doped SWCNTs in our high-throughput reactor, suspended them in DSPE-PEG$_{5k}$ (a biocompatible surfactant coating), and injected small samples into mice. The in vivo specimens were excited at 980 nm and

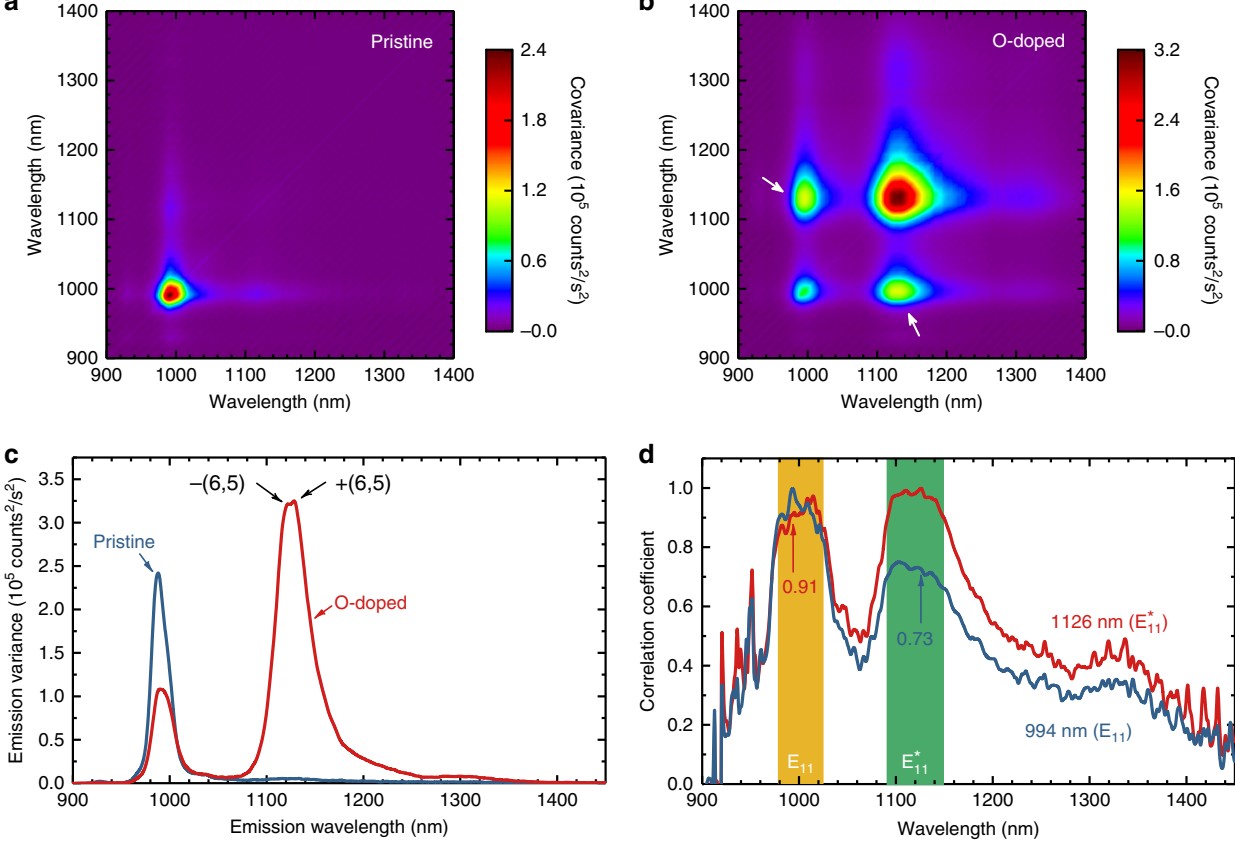

**Fig. 4** Variance spectroscopy of pristine and O-doped (6,5)-SWCNTs. **a** Covariance matrix measured for the (6,5)-SWCNT suspension before treatment. **b** Covariance matrix measured for the (6,5)-SWCNT suspension after treatment. The off-diagonal components (white arrows) reveal the presence of correlated $E_{11}$ and $E_{11}^*$ emissions. **c** Variance spectra (diagonal traces) of frames (**a**) and (**b**). The O-doped trace shows evidence of spectrally resolved $E_{11}^*$ peaks from the two (6,5) enantiomers. **d** Pearson correlation coefficient spectra, at 994 and 1126 nm, of the O-doped (6,5)-SWCNT sample

imaged through optical filters to isolate the $E_{11}^*$ emission. High contrast images displaying clear vascular and lymphatic structure with low autofluorescence backgrounds are shown in Fig. 5c. Note that some organic dyes such as indocyanine green require the blockage of emission wavelengths shorter than 1300 nm to achieve the same level of image contrast because their shorter wavelength excitation leads to much higher autofluorescence backgrounds[41]. Moreover, the dosage used here, only ~100 ng of SWCNTs per mouse (~4 μg kg$^{-1}$), is among the lowest reported for nanoparticle-based fluorescent probes[42]. The ability to locate sentinel nodes is crucial for diagnosing tumor metastasis, studying immune system related disease, and developing immunotherapeutic methods[43]. Our O-doped SWCNTs provide high-resolution imaging of sentinel nodes and therefore can be a new candidate for fluorescence-based lymphoscintography[44] or in vivo lymph node histology.

## Discussion

Table 1 compares different sidewall functionalization methods for creating fluorescent quantum defects in SWCNTs. To date, two main types have been reported: O-doping with retained $sp^2$ hybridization, and organic functionalization giving local $sp^3$ hybridization in the SWCNT. Both product types show similar spectral features and single-photon emission capabilities, although the single-photon emission of O-doped SWCNTs seems more sensitive to the environment[39]. Prior reports of light-assisted reactions to generate SWCNT fluorescent defects have all involved excitation of the nanotubes. By contrast, our method of

photoexciting the reactant precursor gives functionalization rates higher by factors of ~20 to 20,000 than other methods. This photochemical reaction also seems to suppress the introduction of nonfluorescent defects, judging by the lower Raman D/G ratio and absorption perturbation in samples with similarly altered emission spectra.

In conclusion, we have developed a simple and efficient oxygen doping method to create fluorescent quantum defects in SWCNTs using photoexcited NaClO (bleach). This room temperature aqueous reaction takes less than 1 min under 300 nm illumination to reach maximum shift of sample emission to the dopant band. Doping efficiency depends strongly on the identity and concentration of the surfactant used to suspend the nanotubes. The mechanism is proposed to be direct attack on SWCNT side walls by excited O atoms formed through photodissociation of ClO$^-$ ions. Variance spectroscopy shows that most nanotubes in treated samples emit at both the pristine and doped wavelengths, and that only a minority retain pristine emission spectra. Finally, we developed a device allowing larger-scale controlled synthesis of O-doped SWCNTs and demonstrated the effectiveness of the product for high contrast in vivo imaging at SWIR wavelengths.

## Methods

**Sample preparation**. SWCNTs used in this study were grown using the CoMo-CAT and HiPco methods. To prepare a CoMoCAT suspension, the solid material (Aldrich, lot MKBW7869) were added to a 1% aqueous SC (Sigma C1254, lot SLBX2315) solution and dispersed by 1.5 h of active tip-sonication (5 s on/55 s off; Cole-Parmer Ultrasonic Processor) under a 22 °C water bath. The dispersion was then ultracentrifuged (Beckman Coulter SW 32 Ti rotor) at 12,600 g for 3 h

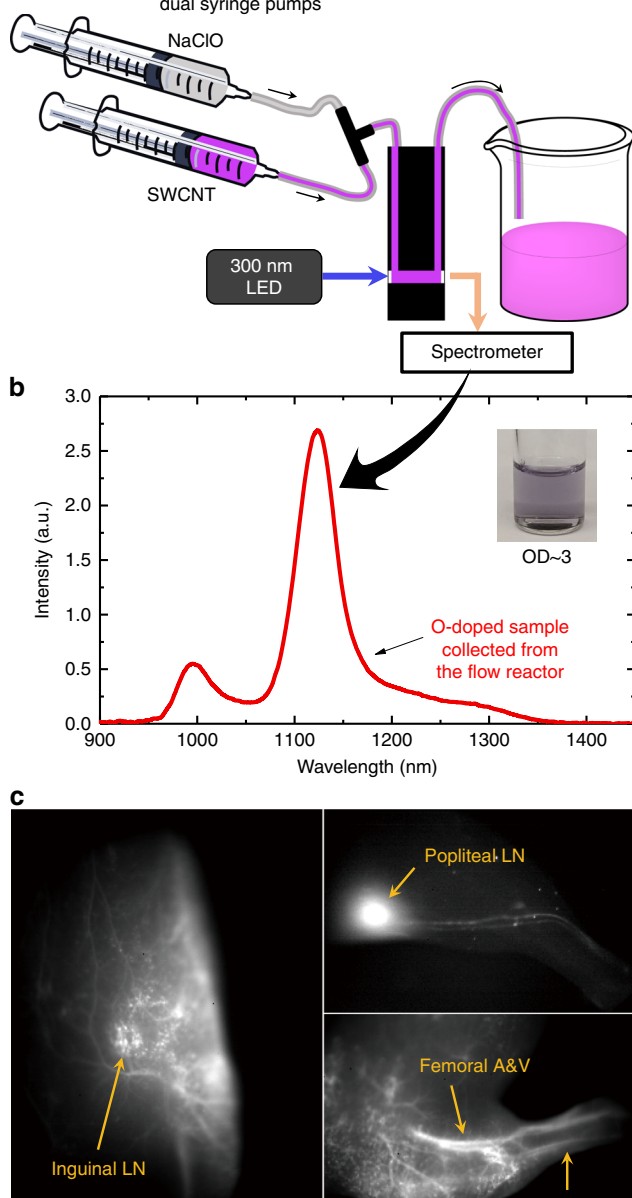

**Fig. 5** Efficient synthesis of O-doped SWCNTs and their application in fluorescence in vivo imaging. **a** Schematic diagram of the flow reactor incorporating syringe pumps and a 300 nm LED for irradiation and fluorescence excitation. Emission was collected through the back cell window and analyzed in a SWIR spectrometer. **b** Emission spectrum of O-doped SWCNTs synthesized in the flow reactor. The solution was diluted by SDC to OD ~0.1 for measurement. (Inset) Photo of O-doped (6,5)-SWCNTs collected directly from the flow reactor. **c** SWIR fluorescence images of O-doped SWCNTs in vivo. Left panel: a black mouse injected intravenously with ~100 ng of O-doped SWCNTs, showing clear inguinal lymph node. Upper right panel: lymphatic drainage after footpad injection (~10 ng) into a white mouse. Lower right panel: clear vasculature imaging from the leg of a black mouse injected with ~100 ng of O-doped SWCNTs. LN lymph node, A&V artery and vein

followed by immediate extraction of the supernatant. Our (6,5)-enriched sample was prepared using a gel separation method modified from that of Wei et al.[45] Two-step instead of multistep elution with various SDC concentrations was used to select racemic (6,5)-SWCNTs. The surfactants were then replaced by 1% SC and the SWCNTs were reconcentrated to an OD of ~4–15 cm$^{-1}$ using tangential flow filtration (mPES/100 kDa, C02-E100–05-N). The HiPco SWCNTs were purchased from NanoIntegris (Batch # HR27-075) and processed using the procedure described above.

**Doping procedure**. The SWCNT suspensions were diluted with water and NaClO to obtain concentrations of 0.035–0.07% SC and ~0.1–1 mM NaClO. For reaction mechanism studies and characterization, we added 300 µL of the prepared solution to a 4 × 10 mm fluorimetric cuvette (Starna Cells 9-Q-10-GL14-S). The cuvette was illuminated at 300 nm at a power density of ~29 mW cm$^{-2}$ for the desired amount of time, usually 40–50 s. SC or SDC surfactants were added to give final concentration near 0.2%. We performed an optional reconcentration step if the SWCNT concentration was too low. For action spectrum measurements, 13 aliquots of (6,5)-enriched SWCNTs in SC and NaClO were prepared for reaction. We then irradiated each aliquot for 50 s with a different wavelength between 250 and 370 nm.

**Optical characterization**. Fluorescence spectra were measured with a NanoLog spectrofluorometer (Horiba) configured with a double excitation monochromator. Sample emission was directed through a 900-nm long-pass filter (Thorlabs FELH0900) followed by a grating monochromator and was then detected by a liquid nitrogen cooled single-element InGaAs detector (Electro-Optical Systems). The spectrofluorometer's excitation system was used separately (normally with a 25-nm band width) for photochemical sample irradiation to induce the oxygen doping reaction. We measured absorption spectra with scanning spectrophotometers (Perkin Elmer Lambda 1050 UV/VIS/NIR or Beckman Coulter DU 800). Raman spectra of SWCNT samples were acquired using a Renishaw inVia™ reflex confocal Raman microscope with 532 nm laser excitation. An objective lens ($f = 15$ mm) focused the laser beam into the liquid sample suspensions, which were adjusted to give absorbance values near 1 per cm at $E_{11}$ peaks. Plotted Raman spectra are the averages of 10 scans from 3100 to 150 cm$^{-1}$. We removed baselines using WiRE software (ver. 4.4).

**Variance spectroscopy**. Variance spectra were measured on a step-scan apparatus described in previous publications[34,35,46–48]. To ensure disaggregation, we tip sonicated the samples at 5 W for 3 min before measurements. A 660-nm diode laser was the excitation source (Power Technologies, Inc.). We acquired 2000 fluorescence spectra at different spatial locations and then processed the data using custom Matlab programs to extract mean, variance, and covariance spectra.

**Single-particle measurements**. SWCNT samples were diluted with 1% SDC solution to desired SWCNT concentrations. Approximately 1 µL of diluted sample was then deposited onto a microscope coverslip. A 40× objective (Zeiss LD C-Apochromat 40×/1.1) in conjunction with a tube lens (Thorlabs TTL200-S8) was used to transmit single particle images to an InGaAs camera (Princeton instruments OMA V-2D). The pixel size was ~500 nm as measured by a resolution test target (Thorlabs R1DS1N). Images were recorded in two wavelength channels, covering 950–1000 nm and 1100–1300 nm, to compare the ratio of the defect or side band emission to the pristine $E_{11}$ emission.

**Theoretical calculations**. Semiempirical methods, mostly PM3, were used for quantum chemical calculations in the HyperChem software package. Energies were determined for optimized structures, if available. For nonequilibrium structures such as "stretched" O–Cl, we used single-point energy calculations. No corrections for configuration interaction were included in the energy calculations.

**In vivo fluorescence imaging**. The O-doped SWCNTs in 1% SC were displaced by DSPE-PEG$_{5k}$ using a modified version of the previously published protocols[49]. In brief, the stock SWCNT dispersion in 1% SC was mixed with an equal volume of ~2 mg mL$^{-1}$ DSPE-PEG$_{5k}$ and dialyzed for 3 days using a 1k MWCO dialysis membrane (Spectra/Por®). After that, the solution was centrifuged at 14,000 rpm (Microfuge® 22R) for 30 min to remove aggregates. The DSPE-PEG$_{5k}$-coated O-doped SWCNTs in 1× phosphate buffered saline were then injected into a mouse. Immediately after injection, we acquired SWIR fluorescence images using a 980 nm diode laser (CNI Laser) for excitation and an InGaAs camera (Princeton Instruments OMA V-2D: 320) for detecting the emission. The excitation intensity at the specimen was limited to ca. 100 mW cm$^{-2}$. An 1150 nm long-pass filter (Thorlabs FELH1150) was placed in the collection path to reject scattered excitation light, and a camera lens (Navitar MVL25M1) and sometimes a separate achromatic doublet (Thorlabs AC508–075-C-ML) were used to focus the image. All in vivo experiments were performed in compliance with the Institutional Animal Care and Use Committee protocols. Animal experiment procedures were preapproved (Protocol #1215–112–18) by the Division of Comparative Medicine and the Committee on Animal Care, Massachusetts Institute of Technology, and in compliance with the Principles of Laboratory Animal Care of the National Institutes of Health, USA.

**Table 1 Comparison of aqueous reactions generating SWCNT fluorescent quantum defects**

| Reference | Defect type | Photoexcited species | D/G Raman ratio | | Relative decrease in $E_{11}$ absorption | $E_{11}^*/E_{11}$ emission ratio | Reaction time (min) |
|---|---|---|---|---|---|---|---|
| | | | Doped | Pristine | | | |
| This work | O-doping | $ClO^-$ | 0.037 | 0.01 | 17% | 5.3 | 0.67 |
| Ghosh et al.[18] | O-doping | SWCNT ($E_{22}$) | 0.17 | 0.03 | 30% | 5.2 | 960 |
| Chiu et al.[19] | O-doping | SWCNT ($E_{22}$) | 0.27 | 0.13 | 9% | 7.7 | 50 |
| Piao et al.[21] | $sp^3$ | – | 0.21 | 0.01 | 24% | 18.1 | 14,400 |
| Kwon et al.[50] | $sp^3$ | – | 0.15 | – | –[a] | 8.9 | 16 |
| Wu et al.[51] | $sp^3$ | SWCNT ($E_{22}$) | 0.04 | 0.016 | – | 1.4 | 30 |

[a]Accurate assessment prevented by background absorption

## Data availability

The authors declare that the data supporting the findings of this study are available within the paper and its Supplementary Information files. All other relevant data that support the findings of this study are available from the corresponding author upon reasonable request.

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

## Acknowledgements
This research was supported by the Koch Institute Frontier Research Program, the Marble Center for Cancer Nanomedicine and a Cancer Center Support (core) Grant #P30-CA14051, and by grants from the National Science Foundation (CHE-1803066) and the Welch Foundation (C-0807).

## Author contributions
C.-W.L. and A.M.B. conceived the idea. C.-W.L. designed the experiments, synthesized the product, characterized the optical properties of the product and conducted the in vivo imaging. Y.Z. measured variance spectra on the products. S.M.B. performed computations to clarify the reaction mechanism and helped interpret results. U.T. helped with intravenous injection and in vivo imaging. S.H. assisted with SWCNT purification and surfactant wash and exchange. R.B.W. and A.M.B. provided project guidance. C.-W.L. and R.B.W. wrote and edited the manuscript. All authors discussed the results and reviewed the manuscript.

## Additional information

**Competing interests:** The authors declare no competing interests.

**Peer Review Information**: *Nature Communications* thanks the anonymous reviewer(s) for their contribution to the peer review of this work. Peer reviewer reports are available.

