## [Peer Review File · Nature Communications]

Reviewers' comments:

Reviewer #1 (Remarks to the Author):

The authors present a new approach for generating carbon nanotube (CNT) oxygen defects, thoroughly characterize the reaction chemistry leading to defect formation and the spectroscopic properties of the functionalized tubes, and demonstrate the potential for the functionalized tubes to act as effective bio-imaging agents. Such defects are gaining increasing attention for the new functionality and potential applications they add for CNTs and are serving as model systems for similar defects being incorporated into other low-D nanomaterials. The approach shown here provides significant advantages over other methods for incorporating oxygen defects, such as ozonation of the tubes. Advantages include harnessing simple and well-defined chemistry, more control over the reaction process and extent of functionalization, more reproducibility in the final result and significant improvement in speed of functionalization. It will be of interest to both the nanotube community and the broader community interested in optical nanomaterials. I can recommend publication in Nature Communications after the following minor concerns have been addressed:

- 1) On page 3, the authors note defects can be either ether-bridged oxygens or organic dopants. Oxygen functionalization can also lead to incorporation of epoxide groups (yielding states that are more red-shifted than E11*), which should be noted. The authors' Science paper (ref. 18) indicates this, as does the work of Ma et al., ACS Nano, 8, 10782 (2014). Both papers indicate that the ether is more thermodynamically favorable, but that the epoxide formation is thermodynamically possible. The spectrum in Fig. 1a furthermore shows features at long wavelengths that could correspond to the epoxide.
- 2) In a related note, on page 7 the authors point to Fig. 2c as giving energies for the reactant and product species in their mechanism. This is presumably for the ether formation and that should be stated more clearly. Additionally, in discussing the quantum chemical calculations behind Fig. 2c, the authors note in the supporting information (page 19 and Table2, page 20) that the epoxide product is roughly equal in energy to the reactants, which is a somewhat different result than comes from calculations done with the ozone-based reaction. Does this result suggest that functionalization using the ClO- photochemistry provides greater selectivity towards ether formation than one gets with the ozonation reaction? This may be an interesting point for the authors to raise.
- 3) On page 4, the authors note the clear appearance of a new (weak) absorbance feature that appears to be the direct optical absorbance associated with the defect state. This is a very nice result, but the authors should note that this is a parallel to the weak absorbance observed previously for sp³ defects by Piao et al (ref. 21). Additionally, while this goes a little beyond the focus of the paper, it would be interesting if the authors could compare the difference in E11* absorbance and emission energies to the reorganization energy results of Kim et al, J. Phys. Chem. C, 120, 11268 (2016). This energy difference defines the vibrational reorganization energy associated with the emission/absorbance processes and is of fundamental interest.
- 4) The authors note on page 9 that their variance data indicates coexistence of defect-state and pristine exciton emission on individual nanotubes. It should be noted this is in agreement with direct imaging results of Hartmann et al, ref. 43. The same is true for their comment on page 10 that doping is not restricted to the tube ends.

Reviewer #2 (Remarks to the Author):

In this manuscript, Belcher et al. report on a new, efficient and rapid process allowing the introduction of fluorescent impurities in semiconducting carbon nanotubes with the objective to provide them with efficient near infrared emissive sites. The mechanism is said to involve photodissociation of ClO⁻ ions in a dilute NaClO/SWCNT solution and the direct attack of nanotube sidewalls by excited O atoms.

The article is well written, clearly illustrated and reports on a novel nanotube functionalization route. The data analysis using a type of fluorescence variance spectroscopy appears to be sound. The article is likely of interest to some of the readers of Nature Communications and may be published after the following comments have been taken into consideration.

Minor comments:

- Title

Replace "Bleach" with "Sodium hypochlorite"

The fact that a cleaning product contains hypochlorite appears to have no significance in this context since the cost of most likely any nanotube derived fluorescent material is determined by the cost for the fabrication of purified nanotube suspensions and not by the chemicals needed for further functionalization.

- Can the authors provide data regarding the lifetime of the excited singlet 1D state of atomic oxygen in aqueous environment? This might support the proposed reaction mechanism.

Point-by-point response to reviewers' comments

Reviewer #1 (Remarks to the Author):

The authors present a new approach for generating carbon nanotube (CNT) oxygen defects, thoroughly characterize the reaction chemistry leading to defect formation and the spectroscopic properties of the functionalized tubes, and demonstrate the potential for the functionalized tubes to act as effective bio-imaging agents. Such defects are gaining increasing attention for the new functionality and potential applications they add for CNTs and are serving as model systems for similar defects being incorporated into other low-D nanomaterials. The approach shown here provides significant advantages over other methods for incorporating oxygen defects, such as ozonation of the tubes. Advantages include harnessing simple and well-defined chemistry, more control over the reaction process and extent of functionalization, more reproducibility in the final result and significant improvement in speed of functionalization. It will be of interest to both the nanotube community and the broader community interested in optical nanomaterials. I can recommend publication in *Nature Communications* after the following minor concerns have been addressed:

1) On page 3, the authors note defects can be either ether-bridged oxygens or organic dopants. Oxygen functionalization can also lead to incorporation of epoxide groups (yielding states that are more red-shifted than E_{11}^*), which should be noted. The authors' *Science* paper (ref. 18) indicates this, as does the work of Ma *et al.*, *ACS Nano*, 8, 10782 (2014). Both papers indicate that the ether is more thermodynamically favorable, but that the epoxide formation is thermodynamically possible. The spectrum in Fig. 1a furthermore shows features at long wavelengths that could correspond to the epoxide.

Response to the comment #1: We thank the reviewer for suggesting consideration of epoxide adducts. We agree that the epoxide is an important possible doping product, and the research of Ma *et al.*¹ have shown clear evidence for this from both experimental measurements and theoretical simulation. We have added text on p. 3 to make contact with this work. The long-wavelength sideband shown in our work is around $1,141\text{ cm}^{-1}$ lower than E_{11}^* . The assigned epoxide-I peak is 175 meV (or $1,411\text{ cm}^{-1}$) lower than E_{11}^* . The major contribution of the sideband might be assigned to the dark K-momentum exciton at the trap site (see Supplementary Fig. 2), but a minor contribution from the epoxide-I cannot be excluded. The following figure shows these relative positions. Therefore, we have also added a sentence on p. 4 to attribute the new sideband as either X_1^* or epoxide defects. Another interesting phenomenon is that in the treated sample, we always found a slight red shift in the E_{11} peak. This might be attributed to the E_{11}^- reported in the work of Ma *et al.*¹

2) *In a related note, on page 7 the authors point to Fig. 2c as giving energies for the reactant and product species in their mechanism. This is presumably for the ether formation and that should be stated more clearly. Additionally, in discussing the quantum chemical calculations behind Fig. 2c, the authors note in the supporting information (page 19 and Table2, page 20) that the epoxide product is roughly equal in energy to the reactants, which is a somewhat different result than comes from calculations done with the ozone-based reaction. Does this result suggest that functionalization using the ClO⁻ photochemistry provides greater selectivity towards ether formation than one gets with the ozonation reaction? This may be an interesting point for the authors to raise.*

Response to the comment #2: (1) We have clarified that Figure 2c represents reaction to form the ether product by adding a sentence on page 7, a sentence in the Fig. 2c figure caption, and showing the computed epoxide energy level separately in Fig. 2c.

(2) More precisely, the total energy of the epoxide product is estimated to be only ca. 3 kcal/mol below that of the reactants (see table below). To further examine the product selectivity, we checked for the epoxide emission features in the spectra of Ghosh *et al.*² The extra sidebands in the range of 1,010 to 1,060 nm appeared in the first 5 hours, which might be from the E_{11}^- or E_{11}^{*+} emissions. But these less-stable forms seem to disappear after 16 hours. The authors attributed this to irreversible photoisomerization into more stable ether form. Therefore, the bulk of the O-SWCNT product apparently ended up in the ether form after some period of irradiation. By comparison, we did not observe significant emission sidebands other than E_{11}^* using the hypochlorite method, and our samples were not irradiated for a long time to allow photoisomerization. Therefore, we conclude that the argument of the higher initial selectivity using hypochlorite method is reasonable. We have added this discussion to p. 22 of the Supplementary Information.

species	stabilization energy (kcal/mol)	
	ether	epoxide
ozone	55	31
hypochlorite	27.37	2.86

3) *On page 4, the authors note the clear appearance of a new (weak) absorbance feature that appears to be the direct optical absorbance associated with the defect state. This is a very nice result, but the authors should note that this is a parallel to the weak absorbance observed previously for sp^3 defects by Piao *et al* (ref. 21). Additionally, while this goes a little beyond the focus of the paper, it would be interesting if the authors could compare the difference in E_{11}^* absorbance and emission energies to the reorganization energy results of Kim *et al*, *J. Phys. Chem. C*, 120, 11268 (2016). This energy difference defines the vibrational reorganization energy associated with the emission/absorbance processes and is of fundamental interest.*

Response to the comment #3: (1) We thank the reviewer for pointing out this reference, and we now cite the information about the prior observation of E_{11}^* absorption for sp^3 defects on page 5.

(2) As for reorganization energies, refer to the figure below, in which G represents the ground state energy surface for the ether-doped SWCNT and X^- is the trapped exciton surface. The simplest energy relation can be written as

$$E_{11}^{*,abs} = \lambda_{X^-} + E_{11}^{*,em} + \lambda_G$$

or

$$E_{11}^{*,abs} - E_{11}^{*,em} = \lambda_{X^-} + \lambda_G .$$

Therefore, the energy difference between absorption and emission equals the total reorganization energy, which is $\lambda_{\text{total}} = \lambda_{X^-} + \lambda_G$. The λ_{total} obtained from this work is ~ 11.9 meV, which is much smaller than the reported calculated λ_G of 70 meV. Dense oxygen doping in our treated sample might result in a reduced reorganization energy, which is also observed in the sp^3 doped samples. We have added the reorganization energy calculation and discussion on p. 8 of the revised Supplementary Information.

3) The authors note on page 9 that their variance data indicates coexistence of defect-state and pristine exciton emission on individual nanotubes. It should be noted this is in agreement with direct imaging results of Hartmann et al, ref. 43. The same is true for their comment on page 10 that doping is not restricted to the tube ends.

Response to the comment #4: We agree and have added sentences to page 9 and page 10 to mention the consistency with the previous study.

Reviewer #2 (Remarks to the Author):

In this manuscript, Belcher et al. report on a new, efficient and rapid process allowing the introduction of fluorescent impurities in semiconducting carbon nanotubes with the objective to provide them with efficient near infrared emissive sites. The mechanism is said to involve photodissociation of ClO⁻ ions in a dilute NaClO/SWCNT solution and the direct attack of nanotube sidewalls by excited O atoms.

The article is well written, clearly illustrated and reports on a novel nanotube functionalization route. The data analysis using a type of fluorescence variance spectroscopy appears to be sound. The article is likely of interest to some of the readers of Nature Communications and may be published after the following comments have been taken into consideration.

Minor comments:

- Title

Replace "Bleach" with "Sodium hypochlorite"

The fact that a cleaning product contains hypochlorite appears to have no significance in this context since the cost of most likely any nanotube derived fluorescent material is determined by the cost for the fabrication of purified nanotube suspensions and not by the chemicals needed for further functionalization.

Response to comment #1: We agree with the reviewer's comment. The "Bleach" in the title has been replaced by "Hypochlorite".

- Can the authors provide data regarding the lifetime of the excited singlet $1D$ state of atomic oxygen in aqueous environment? This might support the proposed reaction mechanism.

Response to comment #2: An isolated singlet oxygen atom $O(^1D)$ has a very long radiative lifetime of ~ 114 s.³ However, in practice its lifetime is far shorter and depends on chemical reactions with its environment. To the best of our knowledge, measurements of the $O(^1D)$ lifetime in aqueous solution have not been reported. Benedikt *et al.*⁴ used plasma-generated oxygen atoms to prove that they are highly stable in aqueous solution, showing no reaction with water, and are only quenched when encountering a reactive species. For example, the authors show an oxygen atom lifetime of 53 ns in 0.5 mM phenol aqueous solution. The 53 ns lifetime represents the mean diffusion time for oxygen atoms to meet a phenol molecule. The lifetime of oxygen in aqueous solution increased greatly to 32 μ s when there was only dissolved O_2 as a quencher. This is consistent with a simulation result, which states that the $O(^3P)$ remains stable in aqueous solution throughout the simulated time scale of 10 ps⁵. The authors also show that $O(^1D)$ forms oxywater ($H_2O\text{--}O$) within the first iteration and remains stable throughout the rest of the simulation⁵. The conversion of oxywater into H_2O_2 was not observed in the simulation, probably due to the energy barrier⁶. Therefore, it is reasonable to suppose that the $O(^1D)$ atoms are stable in water until they reach a reactive species such as SWCNT or O_2 . To further consider our reaction yield, we know that our optimal NaClO concentration is ~ 3 times higher than the concentration of nanotube carbon atoms. We can estimate that the axial spacing between doping sites on a O-SWCNT product nanotube is ~ 100 nm, which corresponds to 8,800 carbon atoms. The NaClO-to-doping site ratio is then 26,000. In other words, we require 26,000 hypochlorite ions to create one ether dopant structure. This low efficiency suggests that most of the $O(^1D)$ atoms are quenched by other reactive species, probably O_2 or surfactants. Therefore only the small fraction of $O(^1D)$ atoms that are formed near nanotube sidewalls can successfully react with SWCNTs. We have added this discussion to p. 23 of the Supplementary Information.

References

1. Ma, X., et al. Electronic structure and chemical nature of oxygen dopant states in carbon nanotubes. *ACS Nano* **8**, 10782-10789 (2014).
2. Ghosh, S., Bachilo, S. M., Simonette, R. A., Beckingham, K. M. & Weisman, R. B. Oxygen doping modifies near-infrared band gaps in fluorescent single-walled carbon nanotubes. *Science* **330**, 1656-1659 (2010).
3. Slanger, T. G. & Copeland, R. A. Energetic oxygen in the upper atmosphere and the laboratory. *Chem. Rev.* **103**, 4731-4766 (2003).
4. Benedikt, J., et al. The fate of plasma-generated oxygen atoms in aqueous solutions: non-equilibrium atmospheric pressure plasmas as an efficient source of atomic $O_{(aq)}$. *Phys. Chem. Chem. Phys.* **20**, 12037-12042 (2018).
5. Verlackt, C. C. W., Neyts, E. C. & Bogaerts, A. Atomic scale behavior of oxygen-based radicals in water. *J. Phys. D: Appl. Phys.* **50**, 11LT01 (2017).
6. Codorniu-Hernández, E., Hall, K. W., Ziemianowicz, D., Carpendale, S. & Kusalik, P. G. Aqueous production of oxygen atoms from hydroxyl radicals. *Phys. Chem. Chem. Phys.* **16**, 26094-26102 (2014).

REVIEWERS' COMMENTS:

Reviewer #1 (Remarks to the Author):

The authors have appropriately addressed all comments. I can recommend publication of the paper in its current form.

Reviewer #2 (Remarks to the Author):

The authors have responded appropriately to all comments by reviewers. The MS can be published as is.